# Methane-Mediated Vapor Transport Growth of Monolayer WSe_2_ Crystals

**DOI:** 10.3390/nano9111642

**Published:** 2019-11-19

**Authors:** Hyeon-Sik Jang, Jae-Young Lim, Seog-Gyun Kang, Sang-Hwa Hyun, Sana Sandhu, Seok-Kyun Son, Jae-Hyun Lee, Dongmok Whang

**Affiliations:** 1School of Advanced Materials Science and Engineering and SKKU Advanced Institute of Nanotechnology (SAINT), Sungkyunkwan University (SKKU), 2066, Seobu-Ro, Jangan-Gu, Suwon-Si, Gyeonggi-Do 16419, Korea; dagu1821@skku.edu (H.-S.J.); limjyyy@skku.edu (J.-Y.L.); suggyoon@skku.edu (S.-G.K.); sanasandhu_7@yahoo.co.in (S.S.); 2Department of Energy Systems Research and Department of Materials Science and Engineering, Ajou University, Suwon, Gyeonggi-Do 16499, Korea; ruche@ajou.ac.kr; 3Department of Physics, Mokpo National University, Muan-gun, Jeollanam-Do 58554, Korea; skson@mokpo.ac.kr

**Keywords:** TMD, 2D material, WSe_2_, monolayer, methane promoter, single-crystal

## Abstract

The electrical and optical properties of semiconducting transition metal dichalcogenides (TMDs) can be tuned by controlling their composition and the number of layers they have. Among various TMDs, the monolayer WSe_2_ has a direct bandgap of 1.65 eV and exhibits p-type or bipolar behavior, depending on the type of contact metal. Despite these promising properties, a lack of efficient large-area production methods for high-quality, uniform WSe_2_ hinders its practical device applications. Various methods have been investigated for the synthesis of large-area monolayer WSe_2_, but the difficulty of precisely controlling solid-state TMD precursors (WO_3_, MoO_3_, Se, and S powders) is a major obstacle to the synthesis of uniform TMD layers. In this work, we outline our success in growing large-area, high-quality, monolayered WSe_2_ by utilizing methane (CH_4_) gas with precisely controlled pressure as a promoter. When compared to the catalytic growth of monolayered WSe_2_ without a gas-phase promoter, the catalytic growth of the monolayered WSe_2_ with a CH_4_ promoter reduced the nucleation density to 1/1000 and increased the grain size of monolayer WSe_2_ up to 100 μm. The significant improvement in the optical properties of the resulting WSe_2_ indicates that CH_4_ is a suitable candidate as a promoter for the synthesis of TMD materials, because it allows accurate gas control.

## 1. Introduction

The discovery of graphene and its unique properties has triggered the development of various types of layered materials [1]. In particular, transition metal dichalcogenides (TMDs), atomically thin semiconductors of the type MX_2_ (M = Mo, W; X = S, Se), have attracted considerable attention as their physical and electrical properties are tunable. Depending on their composition and thickness, two-dimensional (2D) TMDs have a variety of electrical properties ranging from metal, to insulator, to semiconductor, which could lead to a new dimension of atomic thickness for future device applications [2,3]. TMD materials have useful device characteristics, such as a high on/off ratio, a wide range of photoluminescence, and a low subthreshold voltage, making them suitable for spintronics and optoelectronics [4]. Among the numerous TMD materials, WSe_2_ has been extensively studied because its electrical transport properties can be easily adjusted from p-type to bipolar behavior depending on the type of contact metal [5,6,7]. Bulk WSe_2_ crystallizes in the “2H’’ or trigonal prismatic structure (space group *P*6_3_/*mmc*; *a* = 0.330 nm, *c* = 1.298 nm), in which each W atom is surrounded by six Se atoms, defining two triangular prisms. It was also reported that the energy band structure of WSe_2_ can be altered according to its layer number. WSe_2_ shows a direct bandgap of 1.65 eV in the monolayer, compared to an indirect bandgap of 1.2 eV in the multilayered bulk [8,9]. Similar to another 2D layered material, TMD is typically prepared using a mechanical exfoliation method. However, this top-down approach is not suitable for practical high-performance device applications, so bottom-up approaches for large-scale and mass-production have been extensively studied. The chemical vapor deposition (CVD) method is one of the bottom-up approaches that allows the synthesis of large-area TMDs. The CVD growth of TMDs has largely been studied using two different approaches. The first approach is to pre-deposit transition metal sources such as MoO_3_, WO_3_, etc., on the growth substrate and convert them to TMD by sulfidation (or selenization) [10,11,12,13,14,15,16,17,18]. The second is a noncatalytic growth method, in which a transition metal source and sulfur (or selenium) are heat-treated in a growth tube and flowed in a gaseous state to synthesize the TMD layer on a target substrate [19,20]. However, these CVD approaches have not been successful in uniform, high-quality TMD synthesis because it is difficult to control the thickness and nucleation density of TMDs [21]. Recently, to overcome such problems, many researchers have studied various types of promoters and methods applied for CVD-based TMD synthesis to control gas-phase transport of precursors and the reaction of TMD on the growth substrate [22,23,24,25,26]. Ling et al. reported the synthesis of highly- crystalline MoS_2_ at a relatively low growth temperature (650 °C) using various aromatic molecules as seeding promoters [13]. In particular, domain size of MoS_2_ increased up to 60 µm through vaporized aromatic-molecule catalysts such as perylene-3,4,9,10- tetracarboxylic acid tetrapotassium salt (PTAS) and F_16_CuPc. They also reported that uniform monolayer MoS_2_ can be synthesized on the entire area of the SiO_2_/Si substrate; however, the use of such an organic catalyst leaves a residue on the growth substrate that acts as a defect of the synthesized TMD. Another limitation of this method is that it is not applicable to the growth of WS_2_ and WSe_2_, which require high growth temperatures. In addition, inorganic materials were also reported in assisted WSe_2_ growth methods [15,27]. Liu et al. demonstrated a Cu-assisted self-limited growth (CASLG) method that allowed the synthesis of a high-quality, uniform WSe_2_ monolayer while maintaining a balance between high crystallinity and fast growth rates. They explained that Cu atoms, which occupy the hexagonal sites positioned at the center of the six-membered rings of the WSe_2_ surface, induce self-limited growth of WSe_2_ and prevent unwanted reactions [15]. However, this approach also had disadvantages, for example, the synthesized WSe_2_ had small grain sizes with multilayered regions and the vapor pressure of the solid catalyst could not be precisely controlled.

Herein, we report a catalytic growth of the large-area monolayer WSe_2_ by utilizing CH_4_ (methane) with precisely controlled pressure as t promoter. Through a systematic investigation, it is confirmed that grain size and the nucleation density of WSe_2_ can be controlled according to the ratio of carrier gases (Ar/CH_4_). The gas promoter leads to synthesis of about 100 µm size domains of WSe_2_ and significantly reduces nucleation density from 1.6 × 10^5^ to 1.5 × 10^2^ mm^−2^. Various analytical tools such as Raman, photoluminescence (PL), X-ray photoelectron spectroscopy (XPS), and atomic force microscopy (AFM) analysis are used to demonstrate the properties of synthesized monolayer WSe_2_.

## 2. Materials and Methods 

### 2.1. Preparation

The WSe_2_ precursor powders (Alfa Aesar, Ward Hill, MA, U.S., 99.8%; metal basis, 10 microns) were placed on the cleaned alumina boat. Prior to the growth, the SiO_2_/Si wafer (thermal oxide wafer: 300 nm SiO_2_ layer on Si (100), MTI Inc., Richmond, CA, U.S.) substrate was washed by acetone, ethyl alcohol, and deionized (DI) water, for 5 min, to remove the organic residue and was then treated with oxygen plasma (100 sccm, 100 W). High-purity Ar gas (99.999%, JC gas Inc.) and methane gas diluted in Ar (1% CH_4_, 99% Ar, JC gas Inc., Suwon-si, Gyeonggi-do, Korea) were used as carrier gases.

### 2.2. Synthesis of WSe_2_

The homemade CVD system was designed to flow gas in both directions with a three-zone furnace and a double-quartz tube (outer: 34 mm diameter, inner: 15 mm diameter tube). The WSe_2_ powders were placed in an alumina boat located at the center furnace of the homemade CVD. The SiO_2_/Si substrate was cut to 1 cm × 5 cm size and then placed in the left furnace, about 10 cm from the alumina boat. The CVD system was pumped to the base pressure (2 × 10^−3^ torr) by a rotary pump for 10 min and then filled with Ar gas to 760 torr. In the process of increasing the temperature to the WSe_2_ growth temperature, the flow direction of the carrier gas (Ar 200 sccm) was reversed to prevent unwanted deposition. After the temperature reached 1050 °C, the flow direction of the carrier gas was reversed again to allow the evaporated precursor to reach the growth substrate. In the synthesis process, the experiment was carried out by flowing a different ratio of Ar and CH_4_ (1% diluted at Ar) for 60 min at atmospheric pressure. After the reaction, the furnace was quenched down to room temperature while maintaining the gas flow, and the samples were collected for characterization.

### 2.3. Characterzaion of Synthesized WSe_2_

The morphology and size of synthesized WSe_2_ samples were characterized using optical microscopy (OM, Olympus DX51, Tokyo, Japan) and a SEM (JEOL JSM-7401F, JEOL, LTD, Tokyo, Japan) operating at 5 kV and 10 µA. The nucleation density and grain size of WSe_2_ were analyzed using the Image J program tool. The thickness and surface potential of the WSe_2_ monolayer were confirmed by atomic force microscopy and Kelvin probe force microscopy (KPFM) using Park NX10 (Park system, Suwon-si, Gyeonggi-do, Korea) with a Si cantilever Pt-coated tip. X-ray photoelectron spectroscopy analysis was carried out by ESCA2000 spectrometry (Termo Fisher Scientific, Walthan, Massachusetts, U.S.) using monochromatic Al-Kα radiation (1468.6 eV). Raman and photoluminescence spectra were collected with micro-Raman spectroscopy (WITEC Raman system, Ulm, Germany) using a 532 nm laser.

## 3. Results and Discussion

As shown in Figure 1a, monolayer WSe_2_ was synthesized on the SiO_2_/Si substrate by a homemade three-zone furnace CVD using WSe_2_ powder as a precursor. Briefly, the CVD system can control the temperature at each zone and adjust the direction of the carrier gas as required. During the ramping process for increasing the temperature of the furnace, the carrier gas flowed from the right to left direction to prevent the evaporated precursor from reaching the growth substrate, and the flow direction of the carrier gas was reversed during the growth process to synthesize the WSe_2_ monolayer. A 1 × 5 cm^2^ SiO_2_/Si growth substrate was placed 10 cm away from the alumina boat containing the precursor. The growth behavior of WSe_2_ was investigated by observing the product at the same location as the growth substrate, because the morphology and density of the WSe_2_ crystals depended upon the distance between the precursor and the growth substrate [12,16]. Figure 1b illustrates the catalytic growth of WSe_2_ crystals via vapor-solid transport mechanism, when CH_4_ gas diluted in Ar (1% CH_4_, 99% Ar) was used as both a carrier gas and a promoter. Like other catalysts for the growth of 2D materials, such as the Cu substrate commonly used for graphene growth, CH_4_ induces the lateral epitaxy growth of WSe_2_, increasing its grain size while suppressing its vertical growth or deposition. During the synthesis of WSe_2_, methyl radicals and hydrogen are produced by thermal decomposition of CH_4_ at the precursor hot zone (1050 °C) [28]. Methyl radicals can react with oxygen atoms on the SiO_2_ surface to form O-CH_3_, reducing the nucleation site of WSe_2_. In addition, carbon-related radicals can react with the unstable W vapor to form metastable metallo-organic compounds, which may induce growth of low-defect WSe_2_ crystals. Hydrogen is also known to induce the growth of low-defect WSe_2_ crystals while suppressing vertical growth into bilayers and multilayers by etching defective WSe_2_ [29,30,31]. Figure 1c,d show that while randomly distributed triangular WSe_2_ crystals were grown, the size, density, and thickness uniformity of the grown crystal domains varied significantly with or without CH_4_ promoters. When WSe_2_ was grown without CH_4_ gas, grain size of the obtained domains was less than 1 µm and there were many multilayer regions (Figure 1c). However, when CH_4_ gas was used as a promoter, WSe_2_ existed mostly as a monolayer with a grain size of more than 10 µm (Figure 1d). These results clearly show that CH_4_ gas acts as a promoter for the growth of WSe_2_ crystals.

As various parameters affect the CVD growth of TMDs, substrate size, carrier gas velocity, weights of precursor powders, growth time, and characterization regions were set as constant [8,10,13,18,19,22,26]. Based on this, Figure 2a–d show SEM images of WSe_2_ according to the CH_4_ gas ratio. Figure 2a and Appendix A show that the WSe_2_ grain size is less than 1 µm when only Ar gas is used as the carrier gas. Figure 2a and Appendix A also show some parts of the multilayer WSe_2_ regions (dark-colored) with a nucleation density of 1.6 × 10^5^ mm^−2^. By increasing the CH_4_ gas to 50 sccm, the average grain size of WSe_2_ was increased to ~6 µm with a triangular shape and a nucleation density of 5.5 × 10^3^ mm^−2^ (Figure 2b and Appendix A). As the flow of CH_4_ gas was increased to 100 sccm, the synthesized monolayer WSe_2_ showed an average grain size of 9 µm with a nucleation density of 6.8 × 10^2^ mm^−2^ (Figure 2c and Appendix A). Figure 2d and Appendix A show that the domain size of a single crystal monolayer of WSe_2_ increased up to 80 μm when flowing 150 sccm of diluted CH_4_ gas. In this case, the average grain size was 52 μm with a wide distribution due to a lower nucleation density of 156 mm^−2^. From a statistical analysis of domain images in Appendix A, grain size and nucleation density of WSe_2_ were obtained as a function of the CH_4_ gas ratio (Figure 2e and Appendix A). Generally, increasing the CH_4_ gas ratio yielded a lower nucleation density of monolayer WSe_2_ with a larger grain size. The catalytic effect of CH_4_ on the synthesis of large-grain monolayer WSe_2_ was similar to the catalytic growth of other 2D materials (graphene, h-BN, MoS_2_, WSe_2_, etc.) [11,15,32,33,34,35,36].

We also investigated the effects of the CH_4_ promoter on the morphological and optical properties of synthesized WSe_2_ via the nondestructive analysis tools of Raman spectroscopy and PL. Figure 3a,b show the typical Raman mapping (at center wavelength: ~252 cm^−1^) obtained with and without the CH_4_ promoter, respectively. When CH_4_ was used as a carrier gas, the grain size was about 80 µm with a uniform and strong intensity of E^1^_2g_ peak over the synthesized WSe_2_ crystals (Figure 3a). This result is consistent with the SEM results in Figure 2d. On the other hand, when only Ar was used as the carrier gas, the intensities of the measured E^1^_2g_ peaks were much lower and nonuniform (Figure 3b). Figure 3c shows the differences in the typical Raman spectra of WSe_2_ crystals grown with and without a CH_4_ promoter. In the case of CH_4_-assisted growth, Raman peaks corresponding to E^1^_2g_ and A_1g_ modes of single-layered WSe_2_ were observed (Appendix A). When only Ar gas was used, a relatively low E^1^_2g_ peak and an additional small peak at 307 cm^−1^ (corresponding to B^1^_2g_ resonance mode of WSe_2_) were observed. In general, the B^1^_2g_ peak is only active on the bilayer or multilayer of WSe_2_ [5,37]. We also noted that carbon-related Raman signals such as D peak (~1350 cm^−1^), G peak (~1600 cm^−1^), or 2D peak (~2700 cm^−1^) were not observed. These results indicate that CH_4_ acted only as a promoter during WSe_2_ synthesis and did not leave other carbon-related residues. We noted that the WSe_2_ growth temperature (700~750 °C) was too low to form a carbon layer by the reaction of methane on the surface of the SiO_2_/Si substrate [38]. The optical properties of the synthesized WSe_2_ and the effect of the CH_4_ promoter were further investigated using micro-PL with a 532 nm laser.

Figure 3d shows the PL mapping of WSe_2_ synthesized using a CH_4_ promoter (CH_4_:Ar = 150:50). The synthesized WSe_2_ grain exhibited a uniform PL intensity at the 760 nm wavelength, which is equivalent to the PL value measured with exfoliated and synthesized single-crystal monolayer WSe_2_ [5,37,39]. On the other hand, when only Ar (200 sccm) was used as a carrier gas, the PL of synthesized WSe_2_ had a low intensity and showed a wide distribution due to the formation of bilayers and multilayers of WSe_2_, as shown in Figure 3e. The synthesis effects of CH_4_ gas were demonstrated from the representative PL spectrum of each PL mapping shown in Figure 3f. Based on these optical property data, it was confirmed that when using CH_4_ as a promoter in the WSe_2_ growth process, large WSe_2_ grains with uniform monolayers can be synthesized.

As shown in the topology images obtained through tapping mode AFM, the thickness of the synthesized WSe_2_ is uniform to ~0.7 nm, corresponding to the thickness of the monolayer (Figure 4a) [40,41]. A KPFM image of the monolayer WSe_2_ showed a reduction in surface potential of ~300 meV in WSe_2_ due to the electrostatic screening effect and charge distribution of WSe_2_ (Figure 4b) [42]. The work function of the Pt-coated tip was ~4.3 eV, which was obtained by measuring the surface potential of highly oriented pyrolytic graphite (HOPG) (Appendix A). Since the work function of the SiO_2_/Si substrate was 4.6 eV, the work function of the synthesized WSe_2_ was estimated to be ~4.3 eV. This value is equivalent to the work function value of the exfoliated monolayer WSe_2_ [43]. Figure 4c,d show the XPS results of the synthesized monolayer WSe_2_ with four W-4f peaks (W^4+^4f_7/2_: 32.8 eV, W^4+^4f_5/2_: 34.8 eV, W^6+^4f_7/2_: 36 eV, and W^6+^4f_5/2_: 38.2 eV) and two Se-3D peaks (Se 3d_5/2_: 55.1 eV and 3d_3/2_: 55.9 eV). The two W^4+^4f peaks correspond to the binding energy of W bonded to Se atoms, while the two Se-3d peaks point to the binding energy of Se bonded to W atoms. The two W^6+^4f peaks correspond to the binding energy of the W atoms bonded to the O atoms, resulting from the exposure of the synthesized WSe_2_ to air during the XPS analysis. Additionally, there was no W-4f peak at 32 eV and 34 eV, which represent the 1T phase; therefore, it can be confirmed at the WSe_2_ of the 2H phase. These results are consistent with previous reports on WSe_2_ [12,44].

## 4. Conclusions

In summary, we developed a CH_4_-assisted vapor transport growth method to obtain high-quality monolayer WSe_2_ crystals with large domain sizes. Unlike other promoter s or growth promoters previously reported (polymer, halide, and metal), CH_4_ only acts as a promoter for WSe_2_ growth without producing any residue. Moreover, the nucleation density of WSe_2_ was tuned (from 1.6 × 10^5^ to 1.5 × 10^2^ mm^−2^) by using the gas-phase CH_4_ promoter with precise flow control. The characterization of the synthesized monolayer WSe_2_ by Raman, PL, and KPFM confirmed that CH_4_ is a suitable candidate as a promoter for the growth of high-quality monolayer WSe_2_. Finally, our CH_4_-assisted growth approach may be applicable for the controlled growth of high-quality single crystals of other TMDs.

## Figures and Tables

**Figure 1 nanomaterials-09-01642-f001:**
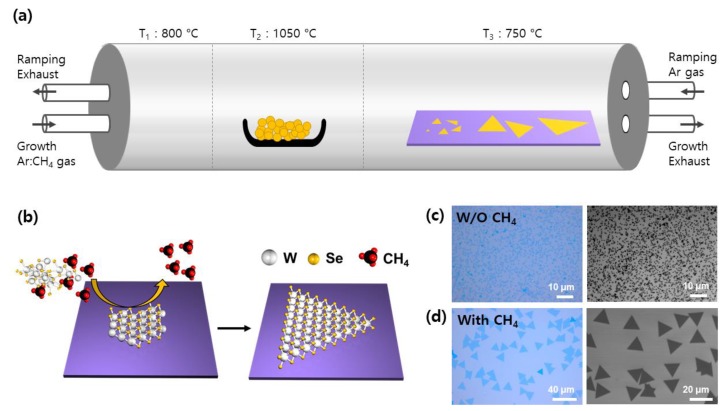
The schematic illustration of methane-mediated WSe_2_ synthesis. (**a**) Sketch of homemade tube-type chemical vapor deposition (CVD) setup. (**b**) Schematic image of WSe_2_ crystal growth by vapor-solid transport mechanism and its growth morphology difference between (**c**) without and (**d**) with methane (CH_4_) gas.

**Figure 2 nanomaterials-09-01642-f002:**
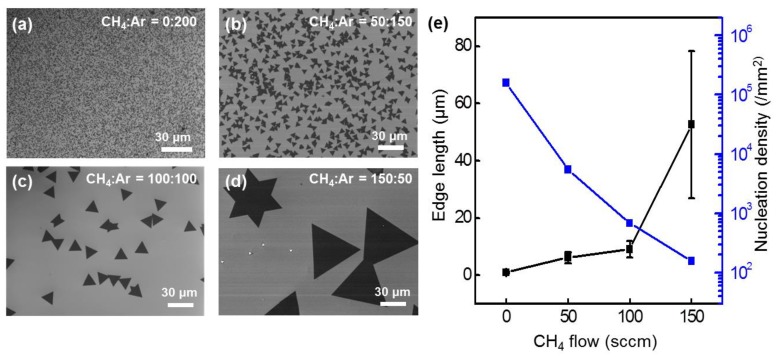
Size control of the WSe_2_ domain by tuning the methane carrier gas ratio. Typical SEM images of WSe_2_ grains synthesized on a SiO_2_/Si substrate with a flow of (**a**) CH_4_:Ar = 0:200, (**b**) CH_4_:Ar = 50:150, (**c**) CH_4_:Ar = 100:100, and (**d**) CH_4_:Ar = 150:50 sccm. (**e**) Edge length (black) and nucleation density (blue) of WSe_2_ domains as a function of the CH_4_ gas ratio. The error bars represent the edge length variations of WSe_2_ domains obtained at the same CH_4_ gas flow.

**Figure 3 nanomaterials-09-01642-f003:**
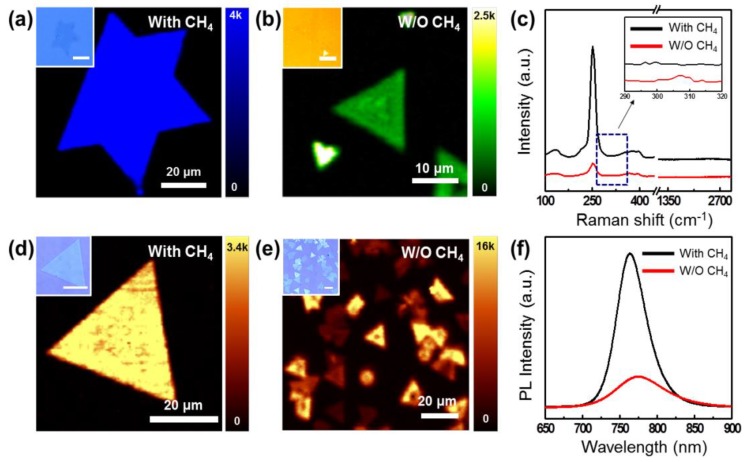
Raman and photoluminescence (PL) investigation of the synthesized WSe_2_ according to the catalytic effects of methane. Raman mapping results of (**a**) CH_4_:Ar = 150:50 sccm, (**b**) Ar gas only as carrier gas, and (**c**) representative Raman spectrum of each mapping result. PL mapping results of (**d**) CH_4_:Ar = 150:50 sccm, (**e**) Ar gas only as carrier gas, And (**f**) representative PL spectrum of each mapping result. Raman and PL results were obtained from a micro-Raman instrument with a wavelength of 532 nm laser. Inset is an OM image corresponding to each mapping region.

**Figure 4 nanomaterials-09-01642-f004:**
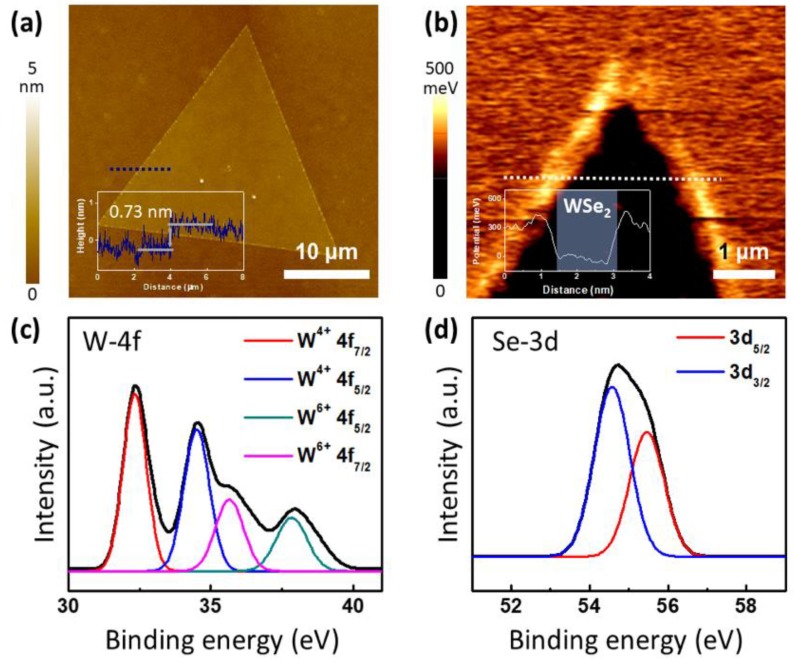
(**a**) Atomice force microscopy (AFM) image of the synthesized monolayer WSe_2_. The inset graph is the height profile corresponding to the blue dot line. (**b**) Surface potential mapping image of WSe_2_ by Kelvin probe force microscopy (KPFM). The inset graph is the surface potential profile corresponding to the white dot line. X-ray photoelectron spectroscopy (XPS) analysis of (**c**) W-4f and (**d**) Se-3d of synthesized WSe_2_.

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
