# Peer review of "Methane-Mediated Vapor Transport Growth of Monolayer WSe2 Crystals"

_nanomaterials, 2019, doi:10.3390/nano9111642_

Round 1

Reviewer 1 Report

The article is devoted to an actively studied topic - the formation of transition element disulfide monolayers. The authors obtained a very interesting result on the increase in the size of the 2D WSe2 islands. The experiments were carried out carefully, the final result is convincingly proven. But, unfortunately, authors only recorded this result, and did not give any explanation for this phenomenon. The role of methane in this phenomenon is not explained at all.

The authors write that methane is a catalyst, but they do not give any explanation why they think so. It is necessary to explain the transport mechanism from the WSe2 source zone. How methane is involved in this? What possible gas species are in the gas phase?

On the other hand, intercalation is characteristic of layered compounds. Maybe methane is introduced between the layers of WSe2. Then in the zone of the starting compound there are intercalation and thermal destruction of WSe2. Thermal destruction of intercalated WSe2 leads to the splitting of the inorganic matrix and allows to obtain particles consisting of 10-20, and maybe single monoatomic layers. Can these particles adsorb methane? When you change the concentration of CH4, the supersaturation changes in the deposition zone, which leads to a change in the density of nucleation. But again, it’s important to at least presume what species are in the gas phase?

It is useful also to compare surface morphology and XRD patters of WSe2 before and after interaction with methane or/and with and without CH4 use. This data is useful to add to the article or the supplement of article. Perhaps these data will help to understand what is happening in the zone of the starting substance.

Now about the use of the term CVD. As you know, in CVD processes, film formation occurs due to chemical reactions. It is clear that at this moment the authors cannot write the reaction sequence. Please, write the total reaction of your process. Evidence that the process goes through a chemical reaction is required. Or is it a sublimation?

Once again, I want to emphasize that a very interesting result was obtained. Discussion of the possible reasons of the detected phenomenon will cause great interest among readers.

The article has a number of technical flaws and misprints, for example:

Line 61. Authors cite articles [22-26] describing catalytic processes. Please, check these references, in these articles there is no description of the catalytical CVD processes, and several do not even describe CVD. The methods of formation of layers that are used in these articles are given below (This information is from these papers).

Ref. 22 - Here we report that the high temperature annealing of a thermally decomposed ammonium thiomolybdate layer in the presence of sulfur can produce large-area MoS2 thin layers with superior electrical performance on insulating substrates.

23 - Here we report that the high-temperature annealing of a thermally decomposed ammonium thiomolybdate layer in the presence of sulfur can produce large-area MoS2 thin layers with superior electrical performance on insulating substrates.

24 - The precursor (ammonium thiomolybdate) together with solvent was transported to graphene surface by a carrier gas at room temperature, which was then followed by post annealing.

25 - Here, we demonstrate the growth of high-quality MS2 (M = Mo, W) monolayers using ambient-pressure chemical vapor deposition (APCVD) with the seeding of perylene-3,4,9,10-tetracarboxylic acid tetrapotassium salt (PTAS).

26 - Here we report a new method for synthesizing high optical quality monolayer MoS2 single crystals up to 25 μm in size on a variety of standard insulating substrates (SiO2, sapphire, and glass) using a catalyst-free vapor-solid growth mechanism.

Line 61. There is no reference on Xi, Li. He is no author of [22-26]. Lines 79-80. Is it proper to call methane a carrier gas? Lines 83 and 89. You use 2 times number 2.1 Lines 89 and 102. Change the name of part 2.3 Line 113. For a better understanding of the paper, it is useful to add the following data to Figure 1:
- indicate the temperatures of all three zones,

- indicate the distance between the alumina boat and the substrate
- the figure shows hydrogen flow, but there is no information on hydrogen in the text of the paper.

Line 116 and 122. It is not clear which substrate you are using. Sometimes you write Si/SiO2, in another place SiO2. If silicon is substrate, please, indicate the thickness of the oxide on the substrate? Line 159. Check out the references. For example, in the article [10] there is no mention of catalytic growth.

Reviewer 2 Report

1) Introduction. Readers could be interested to the crystallographic properties (Space group, lattice parameters ) of WSe2 and to state of Si/SiO2 substrate

2) Line 128: Authors: Like other catalysts for growth of 2D materials, such as Cu substrate  commonly used for graphene growth, CH4 induces the lateral epitaxy growth of WSe2, increasing its grain size while suppressing vertical growth or deposition.

Referee: would you like better explain what do you mean with "lateral epitaxy" in this case ?

3) Figure 2D.  Would you explain ( through a crystallographic reasoning) the superposition of two WSe2 triangular 2D-islands, mutually rotated by 180° ? 

4) Readers could also be interested to know the indices  of the contact plane of the WSe2 triangular 2D-islands on the substrate. 

Round 2

Reviewer 1 Report

Point 1: Line 134: Since the authors do not prove the presence of methane decomposition products in the gas phase, a reference to the article where it is reliably proved is necessary.

Response 1: We are grateful again for the efforts the reviewer #2 has made to review our manuscript. As the reviewer recommended,we have added the related reference in line 135.

  Added the related reference.

- Title of ref. 28: The Thermal Decomposition of Methane

Point 2: The authors suggest only in coverletter (there is no this information in paper), that hexamethyl tungsten (W(CH3)6) is formed. From the literature it is known that this compound is stable only to a temperature of
minus 40 ° C and decomposes above this temperature (“Synthetic Methods of Organometallic and Inorganic Chemistry”, Volume 7, Transition Metals Part 1. 1997, edited by W.A. Herrmann. P.88-89). It is obvious that under the conditions of your experiment the existence of this compound is impossible. Regarding the presence of tungsten vapor, a reference is also needed on their magnitude.

Unfortunately, the authors did not explain the transport mechanism involving methane.

From my point of view, methane can be called a catalyst if you have evidence of an increase in moles of a substance deposited in the presence of methane, compared with the number of WSe2 moles obtained without methane.

Response 2: As you pointed out, hexamethyl tungsten is a very volatile compound. Methyl radicals may react repeatedly with vaporized W to form and decompose as a metastable compounds. We think that the metastable compound, along with the carrier gas, diffuse to the growth substrate and contribute to WSe2 synthesis. We have corrected the methane as a promoter, not a catalyst, for correct representation in the paper. To address this point, we modified it with a promoter instead of the word catalyst in the line below. And, we added words to explain growth mechanism in line 130 follows:

1) Line 130 : via vapor-solid transport mechanism

2) Line 27, 28, 29, 31, 32, 78, 80, 131, 142, 144, 145, 170, 173, 178, 184, 187, 197, 204, 231, 233 and 235: In the revised manuscript, we changed ‘catalyst’ words to ‘promoter’.